

# Analytical model for the power-yaw sensitivity of wind turbines operating in full wake

Jaime Liew [1], Albert M . Urbán [1], and Søren Juhl Andersen [2]

[1]Department of Wind Energy, Technical University of Denmark (DTU), Frederiksborgvej 399, 4000 Roskilde, Denmark
[2]Department of Wind Energy, Technical University of Denmark (DTU), Anker Engelunds Vej 1, 2800 Lyngby, Denmark

**Correspondence:** Jaime Liew (jyli@dtu.dk)

**Abstract.** Wind turbines are designed to align themselves with the incoming wind direction. However, turbines often experience unintentional yaw misalignment, which can significantly reduce the power production. The unintentional yaw misalignment increase for turbines operating in wake of upstream turbines. Here, the combined effects of wakes and yaw misalignment are investigated with the resulting reduction in power production. A model is developed, which considers the trajectory of

each turbine blade element as it passes through the waked wind field in order to determine a power-yaw loss coefficient. The simple model is verified using the HAWC2 aeroelastic code, where wake flow fields have been generated using both medium and high-fidelity computational fluid dynamics simulations. It is demonstrated that the spatial variation of the incoming wind field, due to the presence of wake(s), plays a significant role in the power loss due to yaw misalignment. Results show that disregarding these effects on the power-yaw loss coefficient can yield a 3.5% overestimation in the power production of a

turbine misaligned by $30°$. The presented analysis and model is relevant to low-fidelity wind farm optimization tools, which aim to capture the effects of wake effects and yaw misalignment as well as uncertainty on power output.

## 1  Introduction

As the global wind energy sector continues to grow, there is a strong demand for a decreased levelized cost of energy. With this demand comes an increasing need for accurate and efficient computational tools, which are able to improve the design of wind

farms and optimize annual energy production. In the early phases of wind farm design, optimization tools provide estimates of energy production and the costs during construction, installation or operation. The wind farm planning tools must account for the interactions between nearby wind turbines using wake models. Often, the wake effects and therefore the power production are not accurately modelled when employing engineering wake models and includes substantial uncertainty, see *e.g.* Nygaard (2015) and Peña et al. (2018). Part of the discrepancy and uncertainty might stem from unintentional yaw misalignment (or

yaw error) of turbines inside wind farms. Mikkelsen et al. (2010) reported yaw error on a turbine in freestream wind conditions of up to $20°$ during a measurement campaign of approximately 3 hours. However, McKay et al. (2013) has shown yaw misalignments of up to $35°$ for turbines operating in wake of aligned upstream turbines based on field measurement for a 6 month period. Furthermore, it was show that the yaw misalignment were accentuated further downstream for turbines affected by more wakes. The probability of a wake turbine to be yaw misaligned $\pm25°$ was more than 25% of the time.



Wind turbines which experience yaw misalignment show a reduction in power production. The power-yaw loss function is often expressed as:

$$\frac{P_\gamma}{P_0} = \cos^\alpha \gamma \qquad (1)$$

where $\gamma$ is the yaw misalignment angle between the turbine rotor and the free wind wind, $P_\gamma$ is the power generated by a wind turbine with a yaw misalignment of $\gamma$, and $\alpha$ is the power-yaw loss coefficient. Numerous $\alpha$ values have been proposed in the literature. Based on Blade Element Momentum (BEM) theory, it is commonly concluded that $\alpha = 3$. However, experimental results have often showed that this value overestimate the power loss due to yaw (Aagaard Madsen et al., 2003). Schepers (2001) found experimentally that $\alpha = 1.8$, and Dahlberg and Montgomerie (2005) found a range between $\alpha = 1.88$ and $\alpha = 5.14$. Gebraad et al. (2016) uses a constant $\alpha = 1.88$, determined using the wind farm simulator, SOWFA (Fleming et al., 2013). Medici (2005) found a value of $\alpha = 2$ from wind tunnel data. However, these considerations are simplified and only valid for free-stream conditions. The investigation performed by Urbán et al. (2019) shows that yaw misalignment of a turbine in the wake of another turbine, can exhibit significant variations in the value of $\alpha$. In particular, $\alpha$ depends on the shape of the wake deficit profile, which evolves as it propagates downstream. The wake recovery rate is highly dependent on turbine spacing and ambient turbulence intensity. It was found that $\alpha$ is maximum for a turbine located approximately 4 rotor diameters ($4D$) downstream of another turbine when in a full wake situation. Furthermore, $\alpha$ tends to decrease rapidly as turbine spacing decreases below $4D$, while $\alpha$ converges slowly to a fixed value as turbine spacing increases.

Considering the findings in McKay et al. (2013), the implication of unintentional yaw misalignment can be significant for the total power production of large wind farms. Efforts to reduce the yaw error includes improved measuring techniques for individual turbine control, see *e.g.* Kragh et al. (2013) and Schlipf et al. (2013), as well as farm level control, where the information on wind direction is shared between turbines in close proximity to improve the overall alignment, see Annoni et al. (2019). However, it should be mentioned that part of the yaw misalignment compared to the free stream wind direction should not necessarily be considered a yaw error. The turbines attempts to align itself with the local inflow direction to optimize the power production, where the presence of wake effects may alter the local flow wind direction. Such behaviour was also described by McKay et al. (2013) and shown in Hulsman et al. (2019), where a optimization based on surrogate models showed that the second turbine should indeed align itself with the local wind direction. Archer and Vasel-Be-Hagh (2019) used LES to also show how turbines deep inside the farm could be intentionally yawed for improved performance.

In recent years, there is an increase focus on applying control strategies for both standalone wind turbines and entire wind farms to increase its operational performance. The main focus is generally on power optimization, for example, Knudsen et al. (2015) and Gebraad et al. (2015). A common form of wind farm control for power optimization is wake steering, in which a wake can be redirected away from a downstream turbine by inducing a yaw misalignment in the upstream turbine. Numerous





studies on wake redirection have been performed by Fleming et al. (2016); Gebraad et al. (2016, 2017); Jiménez et al. (2010); Bossanyi (2018); Munters and Meyers (2018), showing improved annual energy production in wind farms ranging between 2% and 8%. These investigations often assume a constant value of $\alpha$ to determine the trade off of directing a wake, with the exception of Munters and Meyers (2018) who modelled the turbines as actuator disks, which could yaw, and Bossanyi (2018),

5    where $\alpha$ is adjusted based on the pitch angle of the yawed turbine. Overlooking the causes and effects of a varying $\alpha$ becomes a problem in the framework of low fidelity wind farm optimisation, where the layout of the wind farm itself can change the values of $\alpha$ for each turbine. For example, both Gebraad et al. (2017); Howland et al. (2019) demonstrate potential power increases in a wind farm by performing wake steering, where the analysis relies on a constant $\alpha$ despite the fact that some turbines are yawed in wake situations.

It is beneficial to further investigate the behaviour of $\alpha$ in order to better predict the trade off of yaw steering, especially in the event that a turbine is yawed when operating in the wake of another turbine. The estimates of these power gain losses could benefit from a more accurate estimation of $\alpha$ by taking into account the increased uncertainty of yaw alignment when a turbine is in a wake.

By overcoming the assumption of a constant $\alpha$, uncertainty in wind farm modelling tools can be decreased. Low fidelity wind farm optimisation frameworks such as TOPFARM (Réthoré et al., 2014), FLORIS (Fleming et al., 2017) or FarmFlow (Soleimanzadeh et al., 2012) could benefit by including the presented model for estimating $\alpha$, allowing for more accurate results. This paper focus on the estimation of power loss of a wind turbine when yawed in wake, and aims to extend the work of

20    Urbán et al. (2019), who used the Dynamic Wake meandering (DWM) model in conjunction with the aeroelastic tool, HAWC2, to study the effects of axisymmetric wake profiles on a misaligned wind turbine. In the presented work, the DWM generated wakes are validated against large eddy simulation (LES) generated wakes. Furthermore, an analytical formulation, based on concepts of blade element momentum (BEM) theory, is presented, which captures the behaviour of $\alpha$ in axisymmetric wake situations. The analytical formulation is able to estimate values of $\alpha$ rapidly, without the need of aeroelastic simulations. The

25    analytical formulation is validated against simulations using the aeroelastic code, HAWC2, where the dynamic wake flow is generated using both DWM and LES. The formulation can be used in existing wind farm optimization frameworks as a power correction for misaligned wind turbines in full wake scenarios.




## 2   Theory

When a downstream turbine in a full wake situation is perfectly aligned with the incoming wind, each blade segment follows a circular trajectory relative to the mean incoming wind direction. For misaligned cases, where the turbine is yawed, each blade segment follows an elliptical path, where the eccentricity of the ellipse increases with yaw angle (Fig. 1a). When these trajectories are plotted on an unfolded polar grid (Fig. 1b), it can be observed that the blade segment passes through different regions of the waked wind field. As yaw angle increases, all blade segments on a rotor experience flow near the wake center for an increasing period of time. This suggests that the spatial distribution of the wake profile could have an effect on how the power output of a turbine changes with yaw angle. It is therefore proposed that the power output contribution of each blade segment depends on the average wind speed experienced as a result of following a trajectory through a nonuniform wind field.

It is convenient to define a transformation between rotor coordinates $(r_R, \psi_R)$ and meteorological coordinates, $(r_m, \psi_m)$ based on the definition of an ellipse as follows:

$$
\begin{bmatrix} r_m \\ \psi_m \end{bmatrix} = \begin{bmatrix} r_R \dfrac{\cos\gamma}{\sqrt{1-\sin^2\gamma\cos^2\psi_R}} \\ \psi_R \end{bmatrix}
\tag{2}
$$

where the semi-major axis of the ellipse is $r_R$ and the eccentricity is $\sin(\gamma)$.

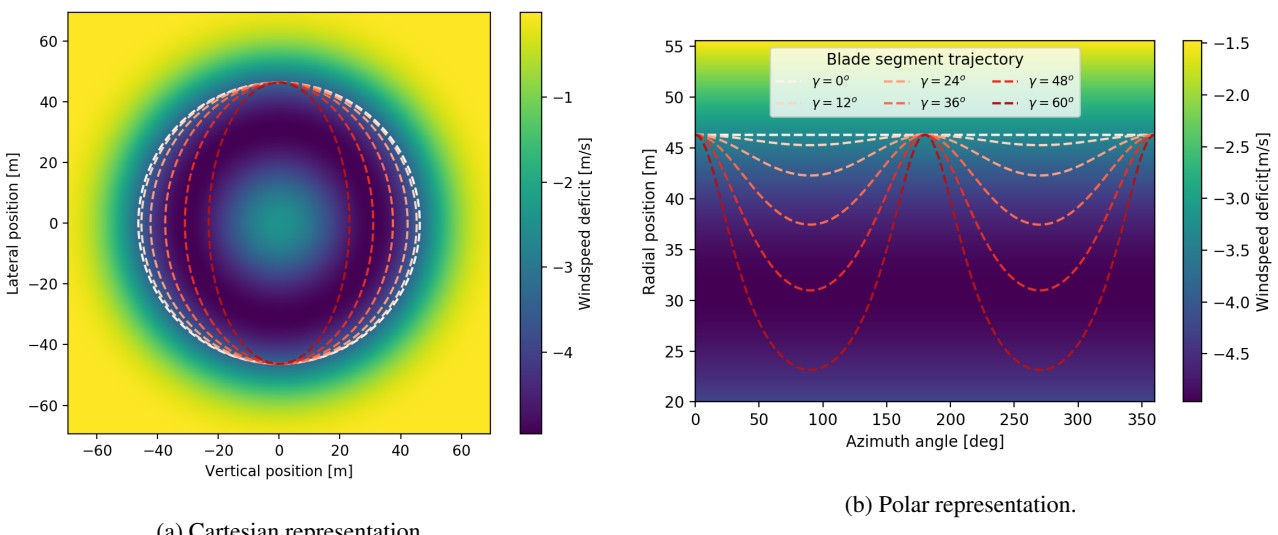

(a) Cartesian representation.

(b) Polar representation.

**Figure 1.** The trajectory of a blade segment close to the tip $(r = 45m)$ through a waked wind field at varying yaw angles, represented in Cartesian coordinates (a) and unfolded polar coordinates (b).





## 2.1 Blade segment effective wind speed in an axisymmetric wake

This section introduces the concept of blade segment effective wind speed. For a blade segment located at radius, $r_R$, the blade segment effective wind speed, $\bar{U}(r_R)$ is defined as the expected value of wind speed experienced by the blade segment as it follows a trajectory through the wind field:

$$\bar{U}(r_R) = \mathbb{E}\left[U(r_m)\right] \tag{3}$$

where $\mathbb{E}\left[.\right]$ is the expected value function, and $U(r_m)$ is assumed to be axisymmetric as displayed in Fig. 1, and is therefore only a function of radius in meteorological coordinates. One way of expressing Eq. (3), assuming the blade segment trajectory is an ellipse as described in Fig. 1, is (Lemma A.2):

$$\bar{U}(r_R) = \mathbb{E}\left[U(r_m)\right] = \underbrace{U(r_R)}_{\text{uniform velocity}} - \underbrace{\int_{r_R\cos\gamma}^{r_R} \frac{dU(\rho)}{d\rho} F_{r_m}(\rho)d\rho}_{\text{added velocity}} \tag{4}$$

where $F_{r_m}(r_m)$ is the cumulative density function of $r_m$ from equation (2) (Lemma A.1):

$$F_{r_m}(r_m) = \begin{cases} 0 & r_m \leq r_R\cos\gamma \\ 1 - \frac{2}{\pi}\arccos\left(\sqrt{\frac{r_m^2 - r_R^2\cos^2\gamma}{r_m^2\sin^2\gamma}}\right) & r_R\cos\gamma < r_m < r_R \\ 1 & r_m \geq r_R \end{cases} \tag{5}$$

From the formulation in (4), it can be observed that the blade segment effective wind speed consists of two additive components. The uniform velocity depends on the wind speed at the rotor radius, whereas an added velocity component depends on radial variations ($dU/dr$) in the wind field. In a uniform wind field, where $dU/dr = 0$, the blade segment effective wind speed remains unchanged when the turbine is yawed. In a nonuniform wind field, the sign of $dU/dr$ determines if the added velocity provides a surplus or a deficit to the blade segment effective wind speed. For instance, Fig. 2 shows the waked wind field for various downstream distances. When the radial wind field function decreases with radius ($dU/dr < 0$), such as in the near wake, the blade segment effective wind speed increases. The opposite occurs when the radial wind field function increases with radius. Therefore, given $U(r_m)$, it is possible to explicitly calculate $\bar{U}(r_R)$ given a value of $\gamma$ and $r_R$ by solving (4).



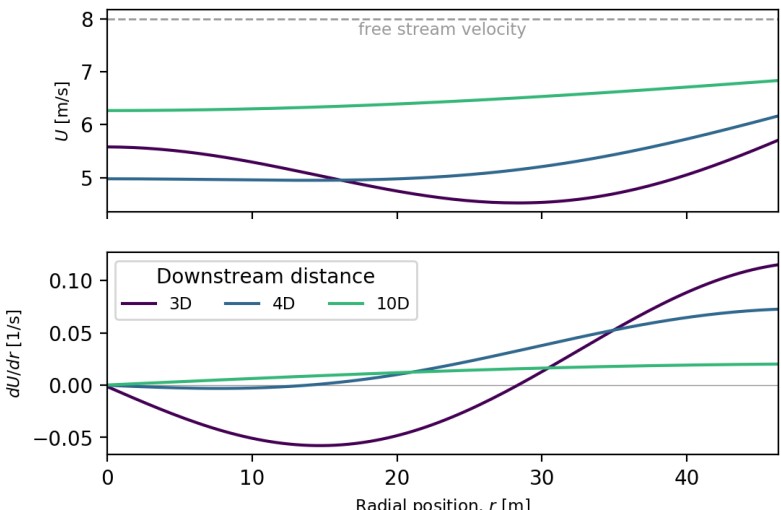

**Figure 2.** Radial functions of wind speed deficit and its derivative for varying downstream distances (generated using DWM).

## 2.2 Modified BEM formulation of wind turbine power in steady yaw and axisymmetric wake

Based of Blade Element Momentum (BEM) theory,

$$\frac{\partial P}{\partial r_R} = A r_R U_\infty^3 \tag{6}$$

where $A = \frac{1}{2}\rho 2\pi 4a(1-a)^2$ is assumed to be constant. In order to stay consistent with the definition of $\alpha$ in (1), as well as the

5    assumption that each blade segment experiences a blade segment effective wind speed, it is proposed that (6) is modified to:

$$\frac{\partial P}{\partial r_R} = A r_R \cos^{\alpha_0} \gamma \bar{U}_\gamma^3(r_R) \tag{7}$$

where $\alpha_0$ is the $\alpha$ fit of (1) for a turbine in free stream conditions. Integrating (7) over the length of the blade gives the total power output of the turbine:

$$P = A \cos^{\alpha_0} \gamma \int_0^R r_R \bar{U}_\gamma^3(r_R) dr_R \tag{8}$$

10    Therefore the power ratio defined on the left hand side of (1) is:

$$\frac{P_\gamma}{P_0} = \frac{\cos^{\alpha_0} \gamma \int_0^R r_R \bar{U}_\gamma^3(r_R) dr_R}{\int_0^R r_R \bar{U}_0^3(r_R) dr_R} \tag{9}$$





It is therefore possible to determine a value of $\alpha$ from (1) which best fits (9) using curve fitting methods. This is achieved in this investigation using a least-squares optimisation:

$$\alpha(D) = \arg\min_{\alpha} \left( \frac{P_{\gamma|D}}{P_{0|D}} - \cos^{\alpha}\gamma \right)^2 \tag{10}$$

where $P_{\gamma|D}$ is the power output if a turbine for a yaw misalignment, $\gamma$ and a downstream distance, $D$. This analytical

approximation of $\alpha$ gives an estimate for a turbine's power sensitivity to yawing while in full wake conditions. The inclusion of $\alpha_0$ in (9) ensures $P_{\gamma}/P_0$ converges to the free stream value as turbine spacing becomes large and wake effects dissipate.

## 3   Method

To determine the value of $\alpha$ for varying downstream distances, four methods are used to estimate the relative power production when a turbine is yawed in a wake situation: (1) aeroelastic simulations with DWM-generated wakes , (2) aeroelastic

simulations with LES-generated wakes (3) analytical model with DWM-generated wakes and (4) analytical model with LES-generated wakes. Each of the model-simulations combinations aim to determine the power output $P_{\gamma|D}$, of the downstream wind turbine with yaw misalignment of $\gamma$ in the full wake of an upstream turbine located distance $D$ apart, as illustrated in Fig. 3.

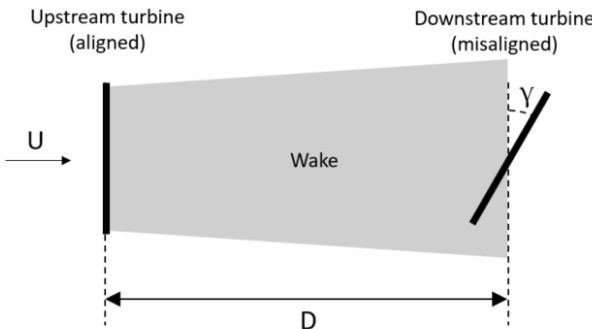

**Figure 3.** Wind turbine layout used in analysis.

To ensure that the combination of wake generation and simulation tools produce comparable results, the free stream wind

speed is fixed at 8m/s with an ambient turbulence intensity of 6% and a shear exponent of 0.14. The free variables, $D$ and $\gamma$, are varied over the ranges of $2D$ to $14D$ and $-30^o$ to $30^o$, respectively. The $\alpha$ coefficient is determined for each downstream distance for the four model-calculation combinations by performing the curve fitting in accordance with the definition of $\alpha$ in Eq. (1).





### 3.1 Aeroelastic Simulation

The aeroelastic simulations (1) and (2) are run using the aeroelastic code HAWC2 (Larsen and Hansen, 2007) using a 2.3MW turbine with a diameter of 96.2m and operating in full wake. The wake generating turbine is similar and has a fixed rotor speed of 1.37 rad/s and blade pitch angle of $-1^o$ to reflect the mean operating conditions at 8m/s, which was previously obtained based

on the flow conditions defined below. The first set uses the DWM model to generate the wake on the target turbine as performed in Urbán et al. (2019). The second set uses a LES generated wake as the input wind field for the aeroelastic simulations which includes the wake dynamics. From the simulations, the mean power output is obtained for different downstream distances and misalignment angles. The results are used to calculate the $\alpha$ coefficient for each downstream distance using (10).

### 3.1.1 DWM wake

The dynamic wake meandering model, as described by Larsen et al. (2008) is used in combination with HAWC2 to mimic the wake effects. The DWM model unifies three key components of wake generation in a computationally efficient manner. These components are the wake deficit profile, the added turbulence profile and wake meandering. The DWM model produces an axisymmetric wake profile using the thrust properties of the upstream turbine. Added wake turbulence is superimposed over the wake profile, and the axisymmetric wake profile is translated to mimic the effects of wake meandering as described in

Madsen et al. (2010), and shown in Fig. 4b(e). The implementation of the DWM model in HAWC2 has been validated against field data in Larsen et al. (2013). Additionally, a steady variation of the DWM model used in Section 3.2.1 has been validated in Keck (2015).

### 3.1.2 LES wake

The turbine and its wake is simulated using the incompressible Navier-Stokes solver EllipSys3D coupled with the aeroelastic

tool Flex5 through the actuator line method. EllipSys3D is based on a finite volume approach with general curvilinear coordinates, see Michelsen (1992) and by Sørensen (1995). The actuator line method as developed by Sørensen and Shen (2002) applies body forces along rotating lines to simulate the presence of the turbine within the flow domain. The position of the rotating lines and applied body forces are determined through the aeroelastic tool, Flex5 by Øye (1996), which gives forces and deflections of the turbine. The effects of atmospheric boundary layer and inflow turbulence are also included using body forces.

The atmospheric boundary layer is modelled with a shear exponent of $0.14$. The inflow turbulence is generated using a Mann box, see Mann (1994) and Mann (1998). The boxes are generated using $\alpha\epsilon^{2/3} = 0.01$, $L = 50$, and $\Gamma = 3.2$, which results in a turbulence intensity of approximately $6\%$. For additional details on the numerical framework, please see Sørensen et al. (2015). The turbine and its wake is simulated in a domain of $20R \times 20R \times 40R$ in the lateral, vertical, and streamwise directions. The turbine is placed in $(10R, 1.3729R, 11R)$ and each blade is resolved by 27 cells. The wind fields consisting of all three velocity

components are extracted for every $1R$ in the wake behind the turbine. These flow fields are used as input to HAWC2 and compared to the wind fields generated using the DWM model as described previously. The wake profiles extracted from the LES framework are expected to be more realistic given that it includes the asymmetric effects of shear on the wake as well as



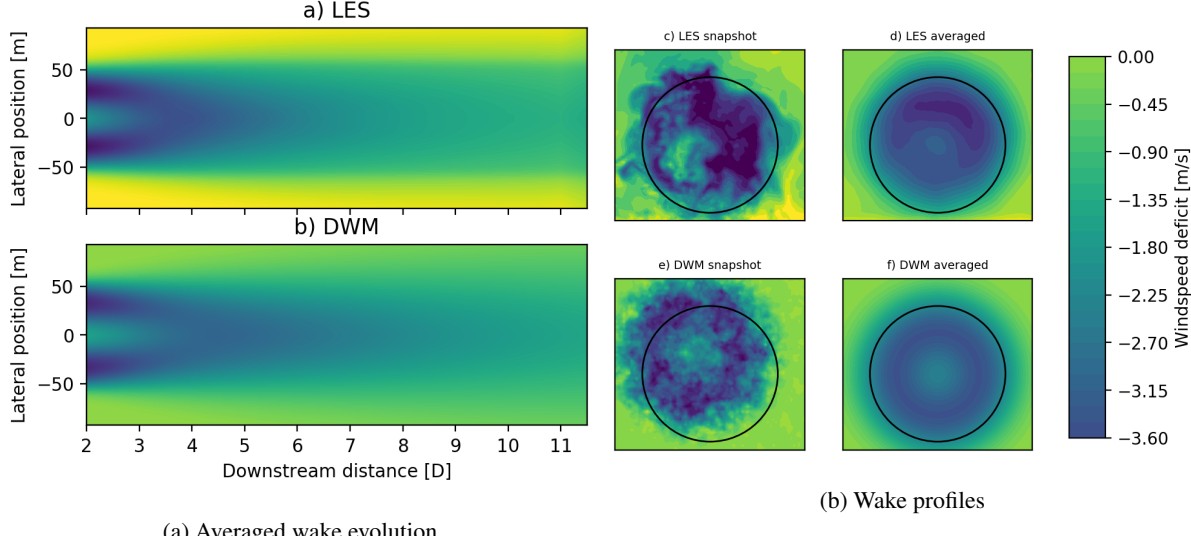

(a) Averaged wake evolution

(b) Wake profiles

**Figure 4.** Visualisations of wake profiles generated using DWM and LES methods.

the nonlinear interactions in a dynamic wake inherent to the flow. These effect are visualized in 4b(c), where the asymmetry and different turbulent structures are more realistic compared the DWM model in 4b(e).

## 3.2 Analytical calculation

Using the wake profiles generated by a standalone version of the DWM model and the time-averaged LES wake deficit profiles,
the analytical formulation described in Section 2 is solved to estimate $\alpha$ without the use of aeroelastic simulations. The radial wind function, $U(r)$ is extracted from the wake profiles at varying downstream distances, shown in 4a. Equation (4) and (9) are solved using numerical differentiation and integration techniques, and the $\alpha$ fit is determined using (10).

### 3.2.1 DWM wake

The DWM model, originally coded within HAWC2, has been externalized for its further use within optimization which results in fast and accurate estimations of a wake profile for a given radial thrust distribution, ambient turbulence intensity and downstream distance (DTU Wind Energy, 2019). A steady wake profile, $U_{DWM}(r)$ is obtained directly from the DWM code. To take into account meandering, $U_{DWM}(r)$ is adjusted by applying a Gaussian smearing using a similar method to Keck (2015),
where the spread of the Gaussian captures the standard deviation of the wake meandering motion.

$$U_{meander}(r_m) = U_{DWM}(r_m) * \left( \frac{1}{\sqrt{2\pi\sigma^2}} e^{-\frac{r^2}{2\sigma^2}} \right) \tag{11}$$





where $*$ is the linear convolution operator. Through a parametric study using the HAWC2 DWM model, the relation $\sigma = 1.493D$ was found to fit best when describing the standard deviation of the wake meandering path. As a result, an axisymmetric, time invariant wake profile is produced (Fig. 4b(f)), which is used in the analytical formulation.

### 3.2.2 LES wake

5     The wake wind field is preconditioned before being used in the analytical formulation by removing the shear profile. This is achieved by subtracting the mean wind field 1D upstream of the wake generating turbine from the downstream wind field.

    Unlike the DWM model, which fully describes the radial wind function, the LES wake at a particular downstream distance is described as a time varying 2 dimensional wind field, $f(x,y,t)$. The mean radial wind speed function is calculated by performing an azimuthal average as:

$$U(r_m) = \frac{1}{NM} \sum_{i=0}^{N} \sum_{j=0}^{M} f(r_m \sin\theta_j, r_m \cos\theta_j, t_i) \qquad \text{where } \theta_j = 2j\pi \tag{12}$$

where $N$ is the number of time steps in the LES wind field, and $N$ is the desired azimuthal discretisation (in this case, $N = 500$). The time- and azimuthally-averaged wake profile, shown in Fig. 4b(d), produces a comparable wake profile to that generated by the DWM model.

## 4   Results

15   Figure 5 presents the power output, normalised with the aligned case, as a function of yaw angle. The expected concave relation between yaw angle and power output is observed and the cosine fit described in (10) is performed. All four methods presented in Fig. 5 present varying curvature of the yaw-power relationship as downstream distance changes. For instance, the difference in curvature can be clearly observed in the LES simulation results (right panels of Fig. 5). The same effect can be observed to a lesser extent for the DWM simulation results on the left of Fig. 5).



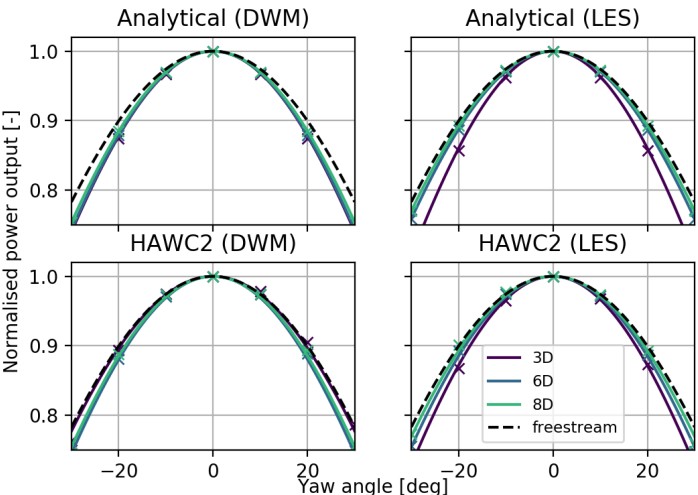

**Figure 5.** Normalised power output as function of yaw angle for different downstream distances. Both calculation methods and wake generation tools are presented.

Although the variations of the power-yaw relation can be observed qualitatively in Fig. 5, it is insufficient at capturing the effect of downstream distance on the power-yaw relation in a quantitative sense. The value of $\alpha$ is therefore presented in Fig. 6 as a function of downstream distance. It is possible to observe that the maximum $\alpha$ value, for both DWM and LES generated wakes, is present at a low turbine spacing between 3D and 5D. As turbine spacing increases, $\alpha$ converges to the free stream

value as the wake dissipates. The free-stream value was found to be 1.7 in the HAWC2 simulations, using both Mann generated and LES generated turbulence fields.

The analytical formulation shows good overall agreement with the aeroelastic simulations. At downstream distances between 5D and 8D, the relative difference between the analytical estimation of $\alpha$ and its respective aeroelastic simulation result is

10 up to 2.7%. For the far-wake region at distances larger than 8D, the analytical model and simulations show a lower relative difference in $\alpha$ of 0.4%. The agreement between aeroelastic simulations and the analytical model is weaker in the near wake scenarios, but the general trend of an increased $\alpha$ is still captured in this region for all four methods.

The maximum value of $\alpha$ at approximately 3D to 4D is due to the strong positive curvature of $U(r)$, leading to a maxi-

15 mum sensitivity to yaw misalignment. As the wake recovers further downstream, this positive slope diminishes, and so $\alpha$ slowly converges to its free stream value. The downstream distance at which $\alpha$ peaks is closely related to the breakdown point, where the wake transitions from near wake to far wake, see Sørensen et al. (2015). In terms of $U(r)$, this point approximately corresponds to the downstream distance at which there is no longer a negative slope.





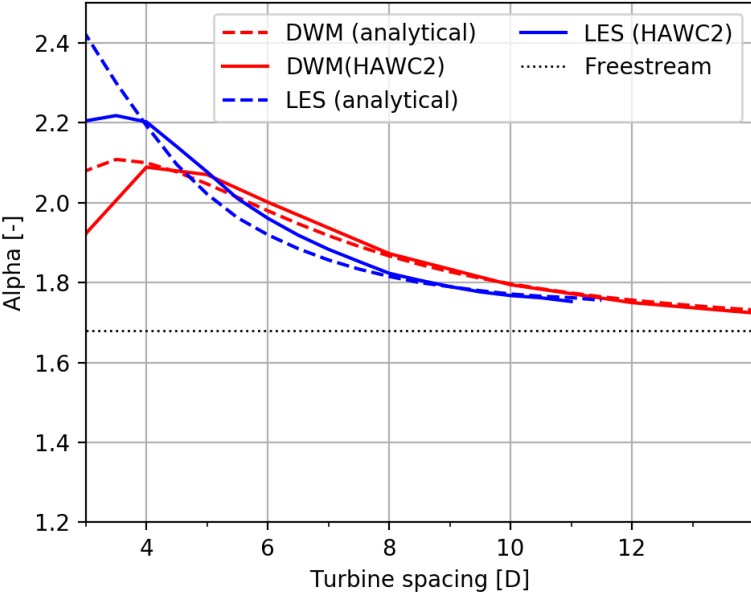

**Figure 6.** $\alpha$ as a function of turbine spacing for the four power calculation methods presented in this investigation.

To highlight the significance of the results in Fig. 6, a wind farm layout consisting of two turbines with a spacing of 6D is considered. Table 1 compares the normalised power output of the downstream turbine using $\alpha = 2.0$ based on the results shown in Fig. 6, and $\alpha = \alpha_0 = 1.7$, which corresponds to the free stream value shown previously. It can be seen that applying the free stream value of $\alpha$ causes an overestimation of the power output when a downstream turbine is yawed, which increases

5   with increasing yaw misalignment. Using typical values of yaw misalignment during wake steering (McKay et al., 2013), it is possible to experience a 3.5% overestimation of the power output for a single turbine at $30^o$ yaw misalignment. Hence, this effect can significantly change the outcome of full wind farm layout optimizations when including the wind direction uncertainty and particular when applying attempting to develop wind farm control including intentionally yaw misaligned turbines in the interior of wind farms.

**Table 1.** Relative power output due to yaw misalignment for $\alpha = 2.0$ and $\alpha = \alpha_0$

| | Relative power $(P_\gamma/P_0)$ | | |
| Yaw misalignment | $\alpha = 1.7$ (free stream) | $\alpha = 2.0$ | difference |
|---|---|---|---|
| $10^o$ | 97.5% | 97.0% | -0.5% |
| $20^o$ | 90.1% | 88.3% | -1.8% |
| $30^o$ | 78.5% | 75.0% | -3.5% |



## 5 Discussion

The estimation of $\alpha$ shows discrepancies in the near wake region depending on the choice of wake generation method (LES or DWM). There are a number of potential sources of this discrepancy. Firstly, the two wake models, although having equal ambient conditions, present slight differences in the rate of mixing due to model differences. For this reason, the break down location of the LES wake appears at a shorter downstream distance than the DWM wake, which explains the $\alpha$ peak occurring at a shorter downstream distance. Secondly, The LES wake is subject to effects not present in the DWM wake, causing differences in the azimuthal and time averaging. These factors include tip vortices, wake rotation and ground effects.

Although the analytical method presented does not consider some physical effects, such as tip losses or rotor induction, the method shows close agreement with aeroelastic simulations in estimating $\alpha$. The results are further reinforced by being able to capture the behaviour of $\alpha$ for both medium and high fidelity wake profile. This provides a correction for which power output can be adjusted for better estimations. It should be noted that the effects of rotor induction on the waked inflow are not considered in the analytical model, HAWC2, as well as both DWM and LES generated wakes.

The investigation is limited to full wake situations, however, by using the azimuthal-time averaging method described in Eq. (12), it is possible to extend the formulation for asymmetric or partial wake cases in future work.

## 6 Conclusions

This paper establishes the link between wake effects and the power sensitivity to yaw misalignment in a wind turbine, quantified by the variable, $\alpha$. A clear trend is found in $\alpha$ through the analysis of HAWC2 aeroelastic simulations using both DWM and LES generated wake flow fields. Namely, $\alpha$ is largest for turbines operating in the near wake region, and $\alpha$ converges to its free stream value as turbine spacing increases. These trends are correctly captured by a theoretical formulation for $\alpha$ presented in this paper. The theoretical formulation correctly anticipates the peak value of $\alpha$, where the wake breaks down, and also converges to the free stream conditions for large downstream distances. The model shows how neglecting the influence of the wake on $\alpha$ can result in power production overestimation up to $3.5\%$ for a yaw misalignment of $30°$.

The simplified model presented in this paper, provides a quick and reliable method to calculate $\alpha$, which can be used for optimisation of wind farm layouts, which includes the uncertainty in the yaw misalignment of wind turbines operating in wake.





**Appendix A: Blade segment effective wind speed derivation**

**Lemma A.1.** *Let $\Psi \in [0, \pi]$ be a uniformly distributed random variable with the cumulative density function,*

$$F_\Psi(\psi) = P(\Psi \leq \psi) = \begin{cases} 0 & \psi \leq 0 \\ \frac{\psi}{\pi} & 0 < \psi < \pi \\ 1 & \psi \geq \pi \end{cases} \tag{A1}$$

*Let $r_m$ be a random variable defined as $r_m = g(\Psi)$ where*

$$g(\Psi) = \frac{R \cos \gamma}{\sqrt{1 - \sin^2 \gamma \cos^2 \Psi}} \tag{A2}$$

*where $R \in \mathbb{R}^+$, $\gamma \in [-\pi, \pi]$. The cumulative distribution function, $F_{r_m}(\rho) = P(r_m \leq \rho)$, of $r_m$ is:*

$$F_{r_m}(\rho) = \begin{cases} 0 & \rho \leq R \cos \gamma \\ 1 - \frac{2}{\pi} \arccos\left(\sqrt{\frac{\rho^2 - R^2 \cos^2 \gamma}{\rho^2 \sin^2 \gamma}}\right) & R \cos \gamma < \rho < R \\ 1 & \rho \geq R \end{cases} \tag{A3}$$

*Proof.*

$$F_{r_m}(\rho) = P(r_m \leq \rho) \tag{A4}$$

$$= P(g(\Psi) \leq \rho) \tag{A5}$$

$$= P\left(\frac{R \cos \gamma}{\sqrt{1 - \sin^2 \gamma \cos^2 \Psi}} \leq \rho\right) \tag{A6}$$

$$= P\left(\cos^2 \Psi \leq \frac{\rho^2 - R^2 \cos^2 \gamma}{\rho^2 \sin^2 \gamma}\right) \tag{A7}$$

$$= P\left(-\sqrt{\frac{\rho^2 - R^2 \cos^2 \gamma}{\rho^2 \sin^2 \gamma}} \leq \cos \Psi \leq \sqrt{\frac{\rho^2 - R^2 \cos^2 \gamma}{\rho^2 \sin^2 \gamma}}\right) \tag{A8}$$

$$= P\left(\arccos\left(-\sqrt{\frac{\rho^2 - R^2 \cos^2 \gamma}{\rho^2 \sin^2 \gamma}}\right) \geq \Psi \geq \arccos\left(\sqrt{\frac{\rho^2 - R^2 \cos^2 \gamma}{\rho^2 \sin^2 \gamma}}\right)\right) \tag{A9}$$

$$= \frac{\arccos\left(-\sqrt{\frac{\rho^2 - R^2 \cos^2 \gamma}{\rho^2 \sin^2 \gamma}}\right) - \arccos\left(\sqrt{\frac{\rho^2 - R^2 \cos^2 \gamma}{\rho^2 \sin^2 \gamma}}\right)}{\pi} \tag{A10}$$

$$= 1 - \frac{2}{\pi} \arccos\left(\sqrt{\frac{\rho^2 - R^2 \cos^2 \gamma}{\rho^2 \sin^2 \gamma}}\right) \tag{A11}$$

$\square$





**Lemma A.2.** *The expected value, $\mathbb{E}[.]$, of a function, $U(r_m)$, is:*

$$\mathbb{E}[U(r_m)] = U(R) - \int\limits_{R\cos\gamma}^{R} \frac{dU(\rho)}{d\rho} F_{r_m}(\rho) d\rho \tag{A12}$$

*Proof.* The expected value of the random variable, $U$ is defined as:

$$\mathbb{E}[U] = \int\limits_{-\infty}^{\infty} U f_U(U) dU \tag{A13}$$

where $f_U(U)$ is the probability density function of $U$. Given that $U$ is a function of radial position, $U(r_m)$, by the law of the unconscious statistician, Eq. (A13) can be written as:

$$\mathbb{E}[U(r_m)] = \int\limits_{r_R\cos\gamma}^{r_R} U(\rho)\frac{dF_{r_m}(\rho)}{d\rho} d\rho \tag{A14}$$

The range of $r_m$ is $(r_R\cos\gamma, r_R)$ as determined from the transformation in Eq. (2), hence the change in the integration limits. Integrating by parts gives:

$$\mathbb{E}[U(r_m)] = U(\rho)F_{r_m}\Big|_{r_R\cos\gamma}^{r_R} - \int\limits_{r_R\cos\gamma}^{r_R} \frac{dU(\rho)}{d\rho} F_{r_m}(\rho) d\rho \tag{A15}$$

$$\mathbb{E}[U(r_m)] = U(r_R) - \int\limits_{r_R\cos\gamma}^{r_R} \frac{dU(\rho)}{d\rho} F_{r_m}(\rho) d\rho \tag{A16}$$

□

*Author contributions.* J.L. developed the theoretical formalism, performed the analytic calculations and processed the aeroelastic simulation
results. A.U. performed the aeroelastic simulations using HAWC2. S.A. generated the LES wake profiles. All authors contributed to the conceptualisation, investigation, and reporting of the research presented in this manuscript.

*Competing interests.* The authors declare that they have no conflict of interest.



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
