# Peer review of "Analytical model for the power-yaw sensitivity of wind turbines operating in full wake"

_Wind Energy Science, 2019_

## Referee Comment (RC1) · Wim Munters (Referee) · 25 Oct 2019

The authors present their manuscript on a novel analytical model for the power-yaw sensitivity of turbines operating in full wake. In this model, they consider the trajectory of each turbine blade element as it passes through the incoming wake, with the aim of analytically determining a power-yaw loss coefficient. The model is validated against HAWC2, with velocity input from both a medium and high fidelity wake model.

The paper is of good quality and I enjoyed reading it. It clearly shows the need of a power-yaw sensitivity model through proper citations to available literature. I believe this work is relevant to the community, as the model can be very useful in increasing the fidelity of engineering models for wind-farm optimization and control.

[Figure]

I do have some comments, which could improve the overall quality of the paper prior to publication. They are listed in the points below. Furthermore, the paper should be thoroughly examined on typos, I've listed some for the first few pages below as well.

**Major comments**

1. p1, l19: *'Part of the discrepancy and uncertainty **might** stem from unintentional yaw misalignment'*. This formulation is quite vague, and should be made more precise if possible. Could the authors make more clear how big they expect / know the effect of yaw misalignment to be? The subsequent citations to literature clearly indicate that yaw misalignments are common, but do not really measure their contribution to the aforementioned discrepancies and uncertainties.

2. It is not completely clear to me why $r_m$ is a random variable, I suppose this is due to uncertainty on $\gamma$. The probabilistic framework introduced in 2.1 hence came a bit as a surprise when reading the text. The source of uncertainty should be stressed more.

3. The yaw misalignment $\gamma$ is introduced on p2, l5 as the yaw misalignment with the free wind direction. Given that the local wind direction can change throughout a wind farm (e.g. due to Coriolis effects), downstream turbines can have a non-zero $\gamma$ whilst still being aligned with the local wind (as the authors also point out on p2. However, does this definition of $\gamma$ then still uphold in the remainder of the text (e.g. Figure 1, and the analytical model), or should $\gamma$ be interpreted as the misalignment to the local wind direction?

4. The main novelty of the current work is being able to account for non-uniform flow conditions in the wake. In a uniform wake, $\bar{U}(r_R) = U(r_R)$, and I suppose from section 2.2 that $\alpha$ would simply equal $\alpha_0$. Is this true? If yes, please mention this. If not, please explain.

5. p 7, l4: Please don't use $D$ both as symbol for the downstream distance. In literature, $D$ virtually always is the turbine diameter. Even in this manuscript, p7 l16 (*ranges of 2D to 14D*), I suspect $D$ indicates the turbine diameter. Use a small $d$ instead to represent the variable downstream distance. A similar thing can be said about the subscript $R$, in the rotor coordinate system $r_R, \psi_R$. $R$ is almost always used to indicate the rotor radius (like you also do in line 29, p8). However, I guess finding another meaningful symbol to represent this is not as simple (since $r$ is already taken for the radial position).

6. Section 3.2.2: The LES wake is post-processed: it is time-averaged, the shear is removed, and azimuthal variations are removed by averaging over the azimuthal direction to obtain a radial wind speed function, which the authors claim to be comparable to that generated by the DWM model. In this sense, I think the added value of including LES in this study should be justified more clearly and perhaps earlier in the manuscript. For instance in Section 5, the authors indicate that wake break down occurs earlier in LES than in DWM. This could probably also be seen from the data in Figure 4, if some cuts at different streamwise locations are taken. I think it would be useful to discuss the relevant observed differences between LES and DWM wakes in Section 3.

7. Added value of LES: Fig2 (why not LES), why even use LES if you make it so close to a DWM? Can the model account for a truly turbulent wake?

8. Figure 5. This is an interesting Figure. Some comments.

   - Would it be insightful to include panel(s) with both the analytical and HAWC2 results on top of each other to highlight the differences?
   - The lines have X markers. I'm assuming this is where the yaw angles have been sampled for the simulations. Please indicate this.

9. Figure 6.

- Perhaps add markers where you sampled these lines (i.e. which simulations you actually ran)
- The differences between your model and HAWC2 seem to increase at lower turbine spacings. However, in practice, such low turbine spacings are rare. Mentioning this in the text would further justify the utility of your model.

10. In the context of wake steering with yaw misalignment, literature has shown that wakes tend to take curled shapes, hence axisymmetric wakes would be rare in farms with active wake steering. Could such wakes also be incorporated using the azimuthal-time averaging? I believe so, and explicitly mentioning this would further strengthen your case.

11. The Lemma's in Appendix A would be better readable if they were self-contained (i.e. defining symbols etc.). Also, in both Lemma's you use captial symbol $R$ again, for a different meaning than simply a subscript indicating the rotor reference frame. Please consider revising this, as this comes off confusing.

12. You propose an analytical method which allows (p 13, l25) *a quick and reliable method to calulate* $\alpha$. Can you compare the computational complexity of your model with HAWC2?

**Minor comments**

1. p2, eq (1): $P_0$ is not defined explicitly, I assume this is simply $P_\gamma$ with $\gamma = 0$ (so a similarly waked turbine with misalignment 0), but in literature $P_0$ is sometimes referring to an unwaked turbine. Better to make this explicit.

2. p4, l10: Mention that $r$ is radial position and $\psi$ is azimuthal angle.

3. p4, Figure 1. This is a nice figure which clearly presents the trajectory of a blade segment. However, in this general theory section, it would be more appropriate to

normalize the length scales in the figure by the rotor radius or diameter, instead of visualizing the concept for a specific turbine with rotor radius 45m. Furthermore, the windspeed deficit could also be normalized with respect to freestream velocity.

4. p5, l9: Indicate clearly that the Lemma is in the appendix. Also, consider changing the order of Lemma A.2 and A.1 (A.2 is referred to earlier in the text than A.1)

5. p6, l4. $\rho$ and $a$ are not defined.

6. Figure 4. The yellow color in the LES flow field is not present in the color scale. Are all figures plotted using the same color scale as shown on the right?

**Typographical**

1. p1, l17: includes include

2. p1, l24: show shown, misalignment misalignments

3. p1, l25: I think it's better to say either 'The probability ... was more than 25%', or 'The waked turbines were yaw misaligned more than 25% of the time', not both.

4. p2, l8: overestimate overestimates

5. p2, l20 includes include

6. p2, l5: wind wind wind

7. p2, l30: increase increased

8. p2, l31: its their

9.  p3, l18: benefit by benefit from

10. p3, l19: focus focuses

11. p3, l24: need of need for

12. p3, l26: optimization optimisation (be consistent in American vs British English)

---

## Referee Comment (RC2) · Anonymous Referee #2 · 4 Nov 2019

article

In this paper, a new analytical model for the estimation of power loss of a wind turbine operating in yaw while being exposed to the wake of an upstream turbine is presented. Here, the power-yaw loss coefficient from the established power-yaw loss function is adapted to match wake inflow conditions in addition to uniform inflows. For this, the path that each blade segment follows through the wake wind field is considered to calculate a blade segment effective wind speed. The model is tested for a wake inflow generated using Large Eddy Simulations (LES) and a wake inflow generated using the dynamic wake meandering (DWM) model, and a validation against aeroelastic simulations is done using the same inflow conditions.

[Figure]

The paper is structured in the following way: An introduction summarizes exist-ing studies of power losses due to yaw misalignment while also pointing out the lack of consideration of wake effects. Then, the theoretical considerations the model is based on are presented, followed by the methods used to test and validate the model, the results of the analysis including an example of the consequences of the mismatch, a discussion and a brief conclusion that summarizes the main findings.

Overall, I think the results presented in this paper are both interesting and rele-vant as they can be used to optimize the procedure of power estimation in wind farms with simplified models, as well as to optimize control algorithms that use wake steering.

Some comments that I believe would improve the overall quality of the paper can be found below. Additionally, while the overall idea of the paper becomes clear and the results are comprehensible, the text would in some sections benefit from being more precise and straightforward, and the text should be checked with respect to language and also typos.

**Major Comments**

1.
*p.1-2:* Here, the terms "power-yaw sensitivity", "power-yaw loss function" and 'power-yaw loss coefficient' are introduced. While the authors are establishing a name for the already existing description, the terminology is only used on the first two pages. I think that if one introduces new terms, one should consistently use them.

2.
*p. 2/ l. 3:* $P_0$ is not introduced, please clarify whether $P_0$ is the power with respect to the free inflow or the inflow of the respective situation where the formula is applied (e.g. wake inflow).

3.

*p. 2/ ll. 14*: First, it is written, that '"The wake recovery rate is highly dependent on turbine spacing and ambient turbulence intensity"' and directly afterwards it is mentioned that '"It was found that $\alpha$ is maximum for a turbine located approximately 4 rotor diameters (4D) downstream of another turbine when in a full wake situation"' - as the ambient conditions determine the wake evolution, it should be pointed out that $\alpha$ is maximum 4D downstream in the situation that was investigated but that this distance might vary depending on the inflow conditions of the upstream turbine.

4.

*p. 3/ ll. 11*: I think that this paragraph needs some work. First, this paragraph is together with the following paragraph used to motivate the necessity of considering a wake inflow for the calculation of $\alpha$. However, the idea is discussed within the framework of wake steering, which is discussed in the previous paragraph. An integration into the previous paragraph could probably help the readability as one aim of an optimized $\alpha$ is a higher precision of the power optimization procedures used for wake steering. Second, the "trade off of wake steering" is a bit vague and also, the aim of wake steering is power production optimization. Third, the term "power gain loss" is not clear to me in this context. Do you mean the trade-off between power losses due to yawing the upstream turbine and the power gain of the downstream turbine?

5.

*p. 4/ l. 4*: $r_R$ is mentioned the first time here but introduced on page 5

6.

*p. 4/ fig. 1*: Was this wind field generated by the DWM model? It should be specified that the inflow conditions of the wake generating turbine are uniform.

[Figure]

7.

*p. 5/ ll. 20:* '"Fig. 2 shows the waked wind field for various downstream distances."' Figure 2 shows the radial variation of the wind speed and its derivative for different downstream positions in a wake generated by the DWM model.

8.

*p. 7:* While D is used as variable for the rotor diameter on page 2, it is used here as variable for the downstream distance. As D is usually used for the rotor diameter, a different variable for the downstream distance should be used.

9.

*p. 7: Methods*: In the introduction, the four different test cases are explained. As the aeroelastic simulations are used to validate the analytical modes, I would mention this again here.

10.

*p. 8/ l. 12:* I would mention that the wake deficit profile generated by the DWM model is depending on the downstream distance.

11.

*p. 10/ l. 11:* '"...where N is the number of time steps in the LES wind field, and N is the desired azimuthal discretization (in this case, N = 500)"' I guess that M is the number of time steps.

12.

*p. 11/ l. 4:* '"... is present at a low turbine spacing between 3D and 5D"' the maximum values are between 3D and 4D.

**Minor Comments**

*p. 1/ll. 19:* '" Part of the discrepancy and uncertainty might stem from unintentional yaw misalignment (or yaw error) of turbines inside wind farms."' This sounds like an assumption of the authors to explain the uncertainty. Connecting this idea with the fact that yaw misalignment occurs regularly (following sentence) could improve the readability: '"As unintentional yaw misalignment (or yaw error) of turbines inside wind farms occurs frequently, this could explain the discrepancy and uncertainty partially. For example, Mikkelsen et al. (2010) reported yaw error on a turbine in freestream wind conditions of up to 20âŮę during a measurement campaign of approximately 3 hours."'

*p. 2/ l. 12::* a new paragraph for the discussion of the power-yaw loss coefficient with respect to wake inflows would emphasize the new focus.

*p. 3/ l. 3:* '"...to determine the trade off of directing a wake"' It would be nice if the sentence was more precise, e.g. '"... to determine the trade-off between power losses due to yawing the upstream turbine and the power gain of the downstream turbine..."'

*p. 3/ l. 5:* '"... is adjusted based on the **blade** pitch angle of the yawed turbine"' - while it should be clear from the context that the blade pitch is meant, I would specify this since new works on floating turbines discuss the pitch, yaw and roll movements of the turbine.

*p. 7/ l. 16:* states that downstream positions between 2D and 14D were investigated, but figure 6 does only show results from 3D downstream.

*p. 8/ll. 5:* '"the first set uses..."', '"the second set uses..."': I would prefer '"simulation"'
over '"set"'.

*p. 8/ll. 7:* '"downstream distance"' and '"turbine spacing"' are used synonymously;
here I would prefer the term '"turbine spacing"' over '"downstream distance"'.

*p. 8, 3.1.2 LES wake:* – several parameters are not introduced ($\alpha$ (this variable should
be renamed), $\epsilon$, L and $\Gamma$, and R)

9
*p. 9, fig. 4*: the color bar depicting the wind speed deficit is incomplete as it lacks the
yellow colors occurring in the LES results.

10
*p. 9 – 3.2 Analytical calculation:*
As in 3.1, it was mentioned that cases (1) and (2) are explained, it should be men-
tioned, that in the following, cases (3) and (4) will be discussed.

11
*p. 11/ l. 14:* '"...is due to the strong positive curvature of U (r)..."' it could be added here
'"is due to the strong positive curvature of U (r) at small turbine spacings (cf. Fig. 2)"'

12
*p. 12/ l. 1:* '"To highlight the significance of the results in Fig. 6, a wind farm layout
consisting of two turbines with a spacing of 6D is considered"' I would probably use

a different formulation, for example "'To give an example of deviations in the power estimation that result from using a constant $\alpha$ as compared to using the new, adapted $\alpha$, a wind farm layout consisting of two turbines with a spacing of 6D is considered"'

13
*p. 13/ ll. 21:* "'theoretical formulation'" - before, "'analytical model'" was used.

**Typographical/Grammar**

If you list several sources, it would be nice to add an "'and'" instead of a "';'" as the separation between the last two sources.

*p. 1/ l. 5:* (+ other positions in the text) "waked" does not exist in this context, instead of "waked wind field", you could use "wake" or in the context of this paper "wake inflow"

*p. 1/l. 2/3:* "'the unintentional yaw misalignment increase**s** for turbines operating in the wakes . . ."'
*p. 1/l.18:* "'Often, the wake effects and therefore the power production are not accurately modeled when employing engineering wake models **which** includes substantial uncertainty"'
*p. 1/ll. 21:* "'However, McKay et al. (2013) ha**ve** shown yaw misalignments of up to 35âŮę for turbines operating in **the** wake**s** of aligned upstream turbines based on field measurement for a 6 month period"'
*p. 2/l. 5:* "'free wind **wind**'"
*p. 3/l. 21:* "'DWM **model**'"
*p. 4/ l. 6:* "'as **the** yaw angle...'"
*p. 6/ l. 2:* "'based **on**'"
*p. 7/ l. 4:* "' is the power output **of** a turbine **for a** yaw misalignment $\gamma$ and a down-

stream distance D'"
*p. 13 / l. 11:* '"high fidelity wake profile**s**'"

---

## Author Comment (AC1) · 10 Dec 2019

**Response to reviewers 1 and 2**

We would like to thank both reviewers for the constructive and detailed comments on our article. The authors have considered the reviewer comments in detail, and we believe that the suggestions have helped strengthen the document before publication. Please find below our responses to your comments. In addition to changes suggested by the reviewers, we have added two additional citations, and minor grammatical changes to the article. Please find attached in the supplementary document a marked-up version showing all changes in the paper.

[Figure]

Yours Sincerely,

Jaime Liew, Albert M. Urbán, and Søren Juhl Andersen

**Reviewer 1**

Major Comments

1. p1, l19: 'Part of the discrepancy and uncertainty might stem from unintentional yaw misalignment'. This formulation is quite vague, and should be made more precise if possible. Could the authors make more clear how big they expect / know the effect of yaw misalignment to be? The subsequent citations to literature clearly indicate that yaw misalignments are common, but do not really measure their contribution to the aforementioned discrepancies and uncertainties.

The reviewer is correct about the vague formulation of the statement present in the paper. However, the authors do not have a precise answer yet regarding the uncertainties of the real implication of the yaw misalignment when wakes are present. This paper needs to be seen as a first step towards that quantification. The intention of this paper is to provide the relevant information and a model to capture such complex phenomena precisely. Next steps are to include measured yaw misalignment in various wind farms to quantify the power loss via our model and compare it with measurements extending the study case for other common inflow cases as partial wakes.

2. It is not completely clear to me why $r_m$ is a random variable, I suppose this is due to uncertainty on $\gamma$. The probabilistic framework introduced in 2.1 hence came a bit as a surprise when reading the text. The source of uncertainty should be stressed
more.

$r_m$ is chosen to be a random variable not because of the uncertainty in $\gamma$, but rather because of the definition of the blade segment effective wind speed, $\bar{U}(r_R)$ in Eq. (3). $\bar{U}(r_R)$ can be thought of as a *probability weighted average wind speed*. In order to determine the weighting, one must know the location of the blade segment in the wind field (determined by the elliptical orbit). It is assumed that the blade segment azimuth angle is uniformly distributed, which leads to the probability distribution of the variable $r_m$, outlined in Appendix A.2. The authors determined that following this line of reasoning is the most straight forward and insightful path to arriving at Eq. 4.

3. The yaw misalignment $\gamma$ is introduced on p2, l5 as the yaw misalignment with the free wind direction. Given that the local wind direction can change throughout a wind farm (e.g. due to Coriolis effects), downstream turbines can have a non- zero $\gamma$ whilst still being aligned with the local wind (as the authors also point out on p2. However, does this definition of $\gamma$ then still uphold in the remainder of the text (e.g. Figure 1, and the analytical model), or should $\gamma$ be interpreted as the misalignment to the local wind direction?

The reviewer raises an interesting concern regarding the definition of yaw mis-alignment. Indeed, the authors are referring to the misalignment between the rotor and the local wind direction. For this reason, the definition of $\gamma$ after Equation 1 has been updated to reflect this.

4. The main novelty of the current work is being able to account for non-uniform flow conditions in the wake. In a uniform wake, $\bar{U}(r_R) = U(r_R)$, and I suppose from section 2.2 that $\alpha$ would simply equal $\alpha$ 0 . Is this true? If yes, please mention this. If not, please explain.

It is indeed true that $\alpha = \alpha_0$ if the wind turbine faces a uniform wind field. This fact has now been mentioned at the end of section 2.2.

5. p 7, l4: Please don't use D both as symbol for the downstream distance. In literature, D virtually always is the turbine diameter. Even in this manuscript, p7 l16 (ranges of 2D to 14D), I suspect D indicates the turbine diameter. Use a small d instead to represent the variable downstream distance. A similar thing can be said about the subscript R, in the rotor coordinate system r R ,$\psi$ R . R is almost always used to indicate the rotor radius (like you also do in line 29, p8). However, I guess finding another meaningful symbol to represent this is not as simple (since r is already taken for the radial position).

The reviewer raises a valid point regarding nomenclature. We have changed the notation for downstream distance to $x$, and have removed all uses of the variable, $R$, so that there is no confusion with the subscript in $r_R$. Section 3.1.2 now refers to distances in terms of rotor diameters instead of rotor radii, and Appendix A.1 has been corrected to not contain any variables with the name $R$.

6. Section 3.2.2: The LES wake is post-processed: it is time-averaged, the shear is removed, and azimuthal variations are removed by averaging over the azimuthal direction to obtain a radial wind speed function, which the authors claim to be comparable to that generated by the DWM model. In this sense, I think the added value of including LES in this study should be justified more clearly and perhaps earlier in the manuscript. For instance in Section 5, the authors indicate that wake break down occurs earlier in LES than in DWM. This could probably also be seen from the data in Figure 4, if some cuts at different streamwise locations are taken. I think it would be useful to discuss the relevant observed differences between LES and DWM wakes in Section 3.

The reviewer raises a good point about providing additional justification of the use of LES. We have extended the first paragraph of the Method section, as well as provide insight into the early breakdown point in section 3.2.2. Please refer to the response of the following question for further remarks.

7. Added value of LES: Fig2 (why not LES), why even use LES if you make it so close to a DWM? Can the model account for a truly turbulent wake?

The purpose of Fig. 2 is to demonstrate a typical wake profile (and its derivative) in the form of a radial function. For this purpose, the choice of DWM or LES in this figure is arbitrary. The use of LES in the paper is not to process the LES to resemble the DWM wake, but rather to have a higher fidelity wake model in the aeroelastic simulations to make up for model limitations of the DWM model. The DWM model is inherently a simplified engineering model, and does not capture the behaviour of a wake in as much detail as LES. The use of both medium and fidelity wake models strengthens the validity of the investigation. Regarding the second question, the strength of the analytical model is that it provides comparable results for $\alpha$ when compared to $\alpha$ determined from turbulent aeroelastic simulations.

8. Figure 5. This is an interesting Figure. Some comments.

- Would it be insightful to include panel(s) with both the analytical and HAWC2 results on top of each other to highlight the differences?

- The lines have X markers. I'm assuming this is where the yaw angles have been sampled for the simulations. Please indicate this.

The authors found that such a figure is not insightful due to overlap of the lines resulting into a crowded figure. It was decided that Fig. 6, showing the curvature of the curves in Fig. 5, is the clearest way to compare the differences in power output due

to yaw in a wake situation. The markers are indeed indications of the sampled points. This has now been indicated in the figure caption.

9. Figure 6.

- Perhaps add markers where you sampled these lines (i.e. which simulations you actually ran)

- The differences between your model and HAWC2 seem to increase at lower turbine spacings. However, in practice, such low turbine spacings are rare. Mentioning this in the text would further justify the utility of your model.

We agree with the reviewer that although the discrepancy between HAWC2 and the model increases at low turbine spacings, these scenarios are rare in practice. We have included this point in paragraph 3 of the Results section. Additionally, we have added markers in Figure 6 indicating the simulations that were run.

10. In the context of wake steering with yaw misalignment, literature has shown that wakes tend to take curled shapes, hence axisymmetric wakes would be rare in farms with active wake steering. Could such wakes also be incorporated using the azimuthal-time averaging? I believe so, and explicitly mentioning this would further strengthen your case.

The reviewer raises an interesting point regarding the use of the analytical model for asymmetric wakes, such as a curled wake. Indeed the azimuthal averaging that we present in 3.2.2 can be applied to a curled wake. This is now mentioned in the last paragraph of Section 5.

11. The Lemma's in Appendix A would be better readable if they were self-contained (i.e. defining symbols etc.). Also, in both Lemma's you use captial symbol

R again, for a different meaning than simply a subscript indicating the rotor reference frame. Please consider revising this, as this comes off confusing.

The reviewer raises an valid concern regarding the Lemmas in Appendix A. The Lemmas have been extended to be self contained, and the use of the variable $R$ has been removed.

12. You propose an analytical method which allows (p 13, l25) a quick and reliable method to calulate $\alpha$. Can you compare the computational complexity of your model with HAWC2?

Thank you for the suggestion. The computational time to determine the power loss exponent using the analytical model is in the order of seconds, whereas stochastic aeroelastic simulations require a time in the order of hours to reliably achieve the same result. We have included this point in the final paragraph of the Introduction.

Minor comments

1. p2, eq (1): $P_0$ is not defined explicitly, I assume this is simply $P_\gamma$ with $\gamma = 0$ (so a similarly waked turbine with misalignment 0), but in literature $P_0$ is sometimes referring to an unwaked turbine. Better to make this explicit.

We agree with the reviewers suggestion, and have defined $P_0$ explicitly after Eq. 1

2. p4, l10: Mention that r is radial position and $\psi$ is azimuthal angle.

$r$ and $\psi$ have now been defined below equation 2.

3. p4, Figure 1. This is a nice figure which clearly presents the trajectory of a blade segment. However, in this general theory section, it would be more appropriate to normalize the length scales in the figure by the rotor radius or diameter, instead of visualizing the concept for a specific turbine with rotor radius 45m. Further- more, the windspeed deficit could also be normalized with respect to freestream velocity.

The reviewer raises a good point regarding the dimensions used in Figure 1. The figure has been adjusted accordingly, and is non-dimensionalized in terms of rotor radius and the free wind speed.

4. p5, l9: Indicate clearly that the Lemma is in the appendix. Also, consider changing the order of Lemma A.2 and A.1 (A.2 is referred to earlier in the text than A.1)

This is a good suggestion. We have now directly referred to Appendix A, and have switched the order of the Lemmas for added clarity.

5. p6, l4. $\rho$ and a are not defined.

Changed accordingly. Thank you.

6. Figure 4. The yellow color in the LES flow field is not present in the color scale. Are all figures plotted using the same color scale as shown on the right?

The color bar has been updated accordingly and the figure layout has been up- dated.

**Typographical**
1. p1, l17: includes include

2. p1, l24: show shown, misalignment misalignments

3. p1, l25: I think it's better to say either 'The probability ... was more than 25%', or 'The waked turbines were yaw misaligned more than 25% of the time', not both.

4. p2, l8: overestimate overestimates

5. p2, l20 includes include

6. p2, l5: wind wind wind

7. p2, l30: increase increased

8. p2, l31: its their

9. p3, l18: benefit by benefit from

10. p3, l19: focus focuses

11. p3, l24: need of need for

12. p3, l26: optimization optimisation (be consistent in American vs British English)

Thank you for the suggestions. The paper has been changed accordingly.

**Reviewer 2**

Major Comments

1. p.1-2: Here, the terms 'power-yaw sensitivity', 'power-yaw loss function' and 'power-yaw loss coefficient' are introduced. While the authors are establishing a name for the already existing description, the terminology is only used on the first two pages. I think that if one introduces new terms, one should consistently use them.

The reviewer raises an important concern regarding the naming convention of

the variable, $\alpha$. The authors have chosen to rename the variable the 'power-yaw loss exponent' to better reflect its behaviour. This terminology has now been included throughout the document.

2. p. 2/ l. 3: $P_0$ is not introduced, please clarify whether $P_0$ is the power with respect to the free inflow or the inflow of the respective situation where the formula is applied (e.g. wake inflow).

We agree with the reviewers suggestion, and have defined $P_0$ explicitly after Eq. 1.

3. p. 2/ ll. 14: First, it is written, that 'The wake recovery rate is highly dependent on turbine spacing and ambient turbulence intensity' and directly afterwards it is mentioned that 'It was found that $\alpha$ is maximum for a turbine located approximately 4 rotor diameters (4D) downstream of another turbine when in a full wake situation' - as the ambient conditions determine the wake evolution, it should be pointed out that $\alpha$ is maximum 4D downstream in the situation that was investigated but that this distance might vary depending on the inflow conditions of the upstream turbine.

The reviewer raises an excellent point regarding the relationship between inflow conditions and the location of the maximum value of $\alpha$. Indeed, the location of the peak is dependent on the atmospheric conditions, especially the turbulence intensity. We have updated the paragraph to make this point clear.

4. p. 3/ ll. 11: I think that this paragraph needs some work. First, this paragraph is together with the following paragraph used to motivate the necessity of considering a wake inflow for the calculation of $\alpha$. However, the idea is discussed within the framework of wake steering, which is discussed in the previous paragraph. An integration into the previous paragraph could probably help the readability as one
aim of an optimized $\alpha$ is a higher precision of the power optimization procedures used for wake steering. Second, the 'trade off of wake steering' is a bit vague and also, the aim of wake steering is power production optimization. Third, the term 'power gain loss' is not clear to me in this context. Do you mean the trade-off between power losses due to yawing the upstream turbine and the power gain of the downstream turbine?

We agree with the reviewer that this paragraph can be integrated with the previous paragraph. This has been done in the revised version. Thank you for bringing our attention to the term 'power gain loss'. This was a typo, and has been corrected to 'power loss'.

5. p. 4/ l. 4: $r_R$ is mentioned the first time here but introduced on page 5

We have now defined $r$ and $\psi$ before Equation (2)

6. p. 4/ fig. 1: Was this wind field generated by the DWM model? It should be specified that the inflow conditions of the wake generating turbine are uniform.

Fig 1 was indeed generated using the DWM model. This has now been mentioned in the caption, as well as its inflow conditions.

7. p. 5/ ll. 20: 'Fig. 2 shows the waked wind field for various downstream distances.' Figure 2 shows the radial variation of the wind speed and its derivative for different downstream positions in a wake generated by the DWM model.

Thank you for the suggestion. The document has been changed accordingly.

8. p. 7: While $D$ is used as variable for the rotor diameter on page 2, it is used

here as variable for the downstream distance. As $D$ is usually used for the rotor diameter, a different variable for the downstream distance should be used.

The reviewer raises a valid point regarding nomenclature. We have changed the notation for downstream distance to $x$.

9. p. 7: Methods: In the introduction, the four different test cases are explained. As the aeroelastic simulations are used to validate the analytical modes, I would mention this again here.

Thank you for the suggestion. The first paragraph of the Method section has been updated accordingly.

10. p. 8/ l. 12: I would mention that the wake deficit profile generated by the DWM model is depending on the downstream distance.

We have specified this now in the text.

11. p. 10/ l. 11: '...where N is the number of time steps in the LES wind field, and N is the desired azimuthal discretization (in this case, N = 500)' I guess that M is the number of time steps.

Thank you for bringing up the typographic error. $M$ is actually the azimuthal discretization, and $N$ is the number of time steps. This sentence has been updated accordingly.

12. p. 11/ l. 4: '... is present at a low turbine spacing between 3D and 5D' the maximum values are between 3D and 4D.

Thank you. Changed accordingly.

Minor Comments

1. p. 1/ll. 19: ' Part of the discrepancy and uncertainty might stem from unintentional yaw misalignment (or yaw error) of turbines inside wind farms.' This sounds like an assumption of the authors to explain the uncertainty. Connecting this idea with the fact that yaw misalignment occurs regularly (following sentence) could improve the readability: 'As unintentional yaw misalignment (or yaw error) of turbines inside wind farms occurs frequently, this could explain the discrepancy and uncertainty partially. For example, Mikkelsen et al. (2010) reported yaw error on a turbine in freestream wind conditions of up to 20$^o$ during a measurement campaign of approximately 3 hours.'

2. p. 2/ l. 12:: a new paragraph for the discussion of the power-yaw loss coefficient with respect to wake inflows would emphasize the new focus.

3. p. 3/ l. 3: '...to determine the trade off of directing a wake' It would be nice if the sentence was more precise, e.g. '... to determine the trade-off between power losses due to yawing the upstream turbine and the power gain of the downstream turbine...'

4. p. 3/ l. 5: '... is adjusted based on the blade pitch angle of the yawed turbine' - while it should be clear from the context that the blade pitch is meant, I would specify this since new works on floating turbines discuss the pitch, yaw and roll movements of the turbine.

5. p. 7/ l. 16: states that downstream positions between 2D and 14D were investigated, but figure 6 does only show results from 3D downstream.

6. p. 8/ll. 5: 'the first set uses...', 'the second set uses...': I would prefer 'simulation'

over 'set'.

7. p. 8/ll. 7: 'downstream distance' and 'turbine spacing' are used synonymously; here I would prefer the term 'turbine spacing' over 'downstream distance'.

9. p. 8, 3.1.2 LES wake: – several parameters are not introduced ($\alpha$ (this variable should be renamed), $\epsilon$, $L$ and $\Gamma$, and $R$) 10. p. 9, fig. 4: the color bar depicting the wind speed deficit is incomplete as it lacks the yellow colors occurring in the LES results.

11. p. 9 – 3.2 Analytical calculation: As in 3.1, it was mentioned that cases (1) and (2) are explained, it should be mentioned, that in the following, cases (3) and (4) will be discussed.

12. p. 11/ l. 14: '...is due to the strong positive curvature of $U(r)$...' it could be added here 'is due to the strong positive curvature of $U(r)$ at small turbine spacings (cf. Fig. 2)'

13. p. 12/ l. 1: "'To highlight the significance of the results in Fig. 6, a wind farm layout consisting of two turbines with a spacing of 6D is considered' I would probably use a different formulation, for example 'To give an example of deviations in the power estimation that result from using a constant $\alpha$ as compared to using the new, adapted $\alpha$, a wind farm layout consisting of two turbines with a spacing of 6D is considered'

14. p. 13/ ll. 21: 'theoretical formulation' - before, 'analytical model' was used.

Thank you for the suggestions. We have taken the suggestions into account and have changed the document accordingly.

Typographical/Grammar

If you list several sources, it would be nice to add an "'and' instead of a "';' as the separation between the last two sources.

p. 1/ l. 5: (+ other positions in the text) "waked" does not exist in this context, instead of "waked wind field", you could use "wake" or in the context of this paper "wake inflow"

p. 1/l. 2/3: "'the unintentional yaw misalignment increases for turbines operating in the wakes . . ."'

p. 1/l.18: 'Often, the wake effects and therefore the power production are not accurately modeled when employing engineering wake models which includes substantial uncertainty'

p. 1/ll. 21: 'However, McKay et al. (2013) have shown yaw misalignments of up to $35^o$ for turbines operating in ĚŽ the wakes of aligned upstream turbines based on field measurement for a 6 month period'

p. 2/l. 5: 'free wind wind'

p. 3/l. 21: 'DWM model'

p. 4/ l. 6: 'as the yaw angle...'

p. 6/ l. 2: 'based on'

p. 7/ l. 4: ' is the power output of a turbine for a yaw misalignment $\gamma$ and a down stream distance $D$'

p. 13 / l. 11: 'high fidelity wake profiles'

Thank you for the suggestions. We have taken the suggestions into account and have changed the document accordingly.

Please also note the supplement to this comment:

[revised manuscript text omitted]

---

## Author Response (AR2)

**Response to Reviewer 2**

**minor comments**

p. 4 - figure (1)
In (a) and (b), the axes are now normalized to the rotor radius (or D/2), and it should be specified in the axes labels. Also, the information that a "trajectory close to the blade tip" was chosen, is missing now in the text and the caption. The velocity deficit is normalized to U_0.

We agree that caption can be made more clear. We have changed the caption to better describe the points brought up by the reviewer.

p. 6 - figure (2)
While the axes in figure (1) are normalized to D/2, here, the radial position is given in m. I would prefer to mention the diameter of the respective turbine, D_Turb=96.2m, once in the text and then normalize all axes for consistency.

The authors agree there should be more consistency regarding the dimensionality of the figures. Figure 2 has now been non dimensionalised.

p. 8 - l. 30 - the footnote for φ looks a bit like an exponent, you could maybe alternatively attach the footnote to "Kolmogorov constant"? 👍

The footnote has now been moved to a different part of the sentence.

p. 10 - figure (4)   plot_wake_evolution_and_snapshot.py
a) and b): the lateral position is again given in m and should be normalized.

The units of the figure have been normalized accordingly.

p. 12 - discussion of figure (6): I think it would be nice to briefly mention here that the results obtained using the two wake models also agree quite well for the far wake in addition to the discussion on the Discussion. 👍

We have extended a sentence on Page 12 line 3 to highlight this point.

p. 13 - l. 25
It might help some readers to explain that a curled wake shape is the result of a yawed turbine. 👍

We have extended a sentence on Page 14 line 16 to better explain the source of a curled wake.

language/grammar
p. 2 - l.29 "...where the second turbine should indeed should..."
p. 3 - l. 27 "...can be performed in a seconds" => a few seconds
p. 6 - formula (6) and l. 4, the "=" was somehow replaced by a "-"**?**
p. 7 - l. 15 "Each of ... aims to ..."
p. 8 - l. 8f. " Simulation (1) uses... Simulation (2) uses"
p. 8 - l. 10f. "different turbine spacings"
p. 9 - l. 3f. "The turbine and its wake are simulated"
p. 9 - l. 7ff "The wake profiles ...given that they includes..."
p. 9 - l. 9 "These effects are..."
p. 11 - caption figure (5) " ... for different turbine spacings"
p. 13, l. 15 "Secondly, The…"

Thank you for the language suggestions. We have changed the manuscript accordingly.

[revised manuscript text omitted]